# Implementation of an International Vessel Wall MR Plaque Imaging Research Network: Experience with the ChAMPION Study

Yannan Yu [1,2,*], Wei-Hai Xu [1], Arindam Rano Chatterjee [3], Todd LeMatty [4], Meng Yao [1], Ming-Li Li [5], Truman Brown [6], Maria Vittoria Spampinato [6], Renee Martin [7], Marc I. Chimowitz [4], Colin Derdeyn [8] and Tanya N. Turan [4,†] on behalf of ChAMPION Investigators

1 State Key Laboratory of Complex Severe and Rare Disease, Department of Neurology, Peking Union Medical College Hospital, Chinese Academy of Medical Sciences and Peking Union Medical College, Beijing 100006, China; xuwh@pumch.cn (W.-H.X.); mengyaosugar@foxmail.com (M.Y.)
2 Department of Radiology, Stanford University, Stanford, CA 94305, USA
3 Mallinckrodt Institute of Radiology, Departments of Neurosurgery and Neurology, Washington University School of Medicine, St. Louis, MO 63130, USA; chatterjee@wustl.edu
4 Department of Neurology, Medical University of South Carolina, Charleston, SC 29245, USA; lematty@musc.edu (T.L.); mchimow@musc.edu (M.I.C.); turan@musc.edu (T.N.T.)
5 Department of Radiology, Peking Union Medical College Hospital, Chinese Academy of Medical Sciences and Peking Union Medical College, Beijing 100006, China; limingli@pumch.cn
6 Department of Radiology, Medical University of South Carolina, Charleston, SC 29245, USA; brotrr@musc.edu (T.B.); spampin@musc.edu (M.V.S.)
7 Department of Public Health Science, Medical University of South Carolina, Charleston, SC 29245, USA; hebertrl@musc.edu
8 Department of Radiology, University of Iowa Carver School of Medicine, Iowa City, IA 52242, USA; colin-derdeyn@uiowa.edu
* Correspondence: yannanyu@stanford.edu or yannanyu0123@gmail.com
† ChAMPION Investigators are indicated in the Acknowledgement section.

**Abstract:** Background and Objective: Intracranial atherosclerosis (ICAS) is one of the most common causes of stroke worldwide. High-resolution Vessel Wall MR imaging (VW-MR) is commonly used to study ICAS, but in order to accelerate advances in the field of VW-MR ICAS research, the establishment of a multicenter research network is needed. We introduce our experience in establishing a collaborative international VW-MR ICAS research network in China and North America using an innovative, disease-specific ICAS imaging phantom for standardization of VW-MR sequences at the sites. Methods: Both the Medical University of South Carolina and Peking Union Medical College functioned as Central Coordinating Centers in the network. PUMC identified research centers within China that had the potential for collaboration on VW-MR ICAS research based on networking and prior experience. All selected centers refined MRI sequences using an ICAS phantom with study principal investigators virtually present in real-time during scanning. MRI sequences were efficiently calibrated utilizing the broad expertise of all members of the research team. All centers further validated MRI sequences with human subjects. Results: We identified 11 Chinese hospitals as the potential collaborating sites for the network. Of the 11 selected sites, six sites were able to complete the required VW-MR scanning and sequence refinement using the ICAS phantom and subsequent human subjects. Conclusion: The study demonstrated the feasibility of establishing a cross-continent collaborative VW-MR research network and the use of a disease-specific phantom to facilitate convenient and efficient sequence modification for image quality standardization, which is needed for future multicenter VW-MR studies.

**Keywords:** intracranial atherosclerosis; vessel wall MR imaging; international collaboration; research network

## 1. Introduction

Intracranial atherosclerosis (ICAS) is one of the most common causes of stroke worldwide [1]. Stroke is the leading cause of death in China and is responsible for 1.6 million deaths annually [2]. There are an estimated 7.5 million stroke survivors, with 2.5 million new stroke cases per year [2]. ICAS, the leading cause of stroke in China, causes approximately 1.2 million of these strokes per year (46.6%) [3]. In addition to stroke patients with ICAS, up to 13–15% of people in China may have asymptomatic ICAS that has not yet become clinically apparent [4]. With a population of almost 1.4 billion people, that means that up to 210 million people may be at risk of stroke due to ICAS. The significance of this high prevalence is two-fold: (1) A large number of patients would benefit from the identification of predictors of stroke risk, particularly if this led to new preventative treatments, and (2) The existence of a large number of affected patients provides a large pool from which to recruit ICAS subjects for studies that would identify predictors of stroke in ICAS and test new therapies. Assessing the risk of ICAS plaque imaging biomarkers requires recruitment from a large pool of subjects. For example, to validate an imaging biomarker of intraplaque hemorrhage (IPH) as a surrogate marker for recurrent stroke, based on an estimated IPH frequency of 20% in symptomatic and 5% in asymptomatic patients [5], if the 1-year recurrent stroke risks were 30% and 10% respectively, we would need to follow 270 patients to detect a difference between IPH+ and IPH− patients (2-sided test, 90% power). After controlling for other plaque features and adjusting for risk factor control, the number of patients needed to assess one plaque feature would be even higher. With multiple potential plaque features to study, the high prevalence of ICAS in China makes research in this population critical for imaging biomarker validation as well as the development of new prevention and treatment strategies.

High-resolution Vessel Wall MR imaging (VW-MR) is increasingly being validated for the evaluation of ICAS plaque features [5–11]. The generalizability of these research findings is limited by a lack of VW-MR protocol standards, but consensus recommendations are emerging for specific VW-MR imaging protocols [12]. An efficient method for standardization of accepted VW-MR imaging sequences across sites is critical for the success of ICAS multi-center clinical trials.

China was perfectly poised to gain from further development of a VW-MR ICAS research network. While having an abundance of 3 Tesla (3T) MRI scanners, China lacked the robust collaborative network required to implement VW-MR protocols consistently across multiple research centers and a mechanism for ensuring image quality. This project provided an opportunity for Chinese researchers to design a platform for multicenter investigator-initiated ICAS research utilizing the support and guidance of experienced international collaborators. The goal of the China-America MRI plaque imaging and outcome network (ChAMPION) initiative was to determine the feasibility of VW-MR sequence standardization at centers in the USA and China using a disease-specific ICAS phantom containing individual plaque components (lipid core and fibrous cap) [13].

## 2. Methods

### 2.1. Overview of Project

The investigators at the Medical University of South Carolina (MUSC) [6,14–16] and Peking Union Medical College (PUMC) [5,8,11,17–20] functioned as the Central Coordinating Centers (CCC) in the ChAMPION network over two years, leading a group of sites within their respective regions. Research work derived from US sites has been published previously [13]. Given that MUSC already had experience in establishing a VW-MR network using a disease-specific ICAS phantom, MUSC helped guide the development of the Chinese network. Therefore, this project emphasized the development of the Chinese CCC and strengthened the collaborations between the research groups.

ChAMPION was led by an Executive Committee, composed of the study Co-Principal Investigators, Neuroradiology and Neurology Co-Investigators, and Project Managers, all of whom were involved in VW-MR ICAS research previously. MUSC and PUMC

worked collaboratively to select sites for the network. An external Advisory Committee consisting of experienced ICAS clinical trialists provided guidance on the implementation of the network.

### 2.2. Development of PUMC as a Central Coordinating Center

The first step in network development was establishing the infrastructure support for PUMC. A Project Manager was hired to assist the PI with the regulatory and administrative responsibilities at PUMC. The Project Manager served as the liaison between the Chinese sites and the CCC for the purposes of data and image collection. The PUMC CCC oversaw the development of any necessary contracts (related to site payments and intellectual property) with the Chinese sites. The CCC ensured the protection of human subjects from research risks by monitoring the regulatory process at each site according to international Human Subjects regulations. The PUMC PI and Project Manager performed site visits to meet with study personnel, assist with the implementation of the MRI sequences, and ensure the protocols were being implemented properly.

### 2.3. Selection of Sites in China

The PUMC PI identified 10–15 centers within China that had the potential for collaboration on VW-MR ICAS research based on networking and prior experience. PUMC initiated contact with candidate institutions that expressed interest in collaborating and collected information regarding suitability as a site using case report forms that were developed by the ChAMPION Executive Committee (see Supplemental Material). This data included (i) number of ICAS patients, (ii) availability of 3T MRI, (iii) history of effective implementation of research protocols, and (iv) existing research personnel and capacity to implement a VW-MR research protocol. After data collection was completed, the ChAMPION Executive Committee evaluated the centers. Eleven sites were identified as key potential collaborators based on scoring highly in the criteria listed above and strategic geographic location within China to provide geographic and ethnic variation within the network.

### 2.4. Development of VW-MR Research Infrastructure

#### 2.4.1. Selection of Imaging Sequences

Given the MUSC experience with VW-MR sequence refinement using the basilar ICAS phantom and human subjects, the MUSC protocols were used as the initial starting point for further refinement. The sequences were performed on a 3T MRI with a 32-channel head coil with the following single slab 3D acquisitions (total scan time 40 min): T2-weighted: TR/TE 1500/66, matrix $256 \times 256$, 11 slices, thickness 1.2 mm, FOV 104 mm, FA $180°$; Fluid attenuated inversion recovery (FLAIR): TR/TE 2500/14, inversion time 1069, matrix $256 \times 197$, 11 slices, thickness 1.2 mm, FOV 100 mm, FA $140°$; 3D TOF angiography for localization of the artery (human subjects only); T1-weighted: TR/TE 458/16, matrix $320 \times 320$, 11 slices, thickness 1.2 mm, FOV 128 mm, FA $180°$, pre-contrast and post-contrast (human subjects only).

#### 2.4.2. Image Quality Assessment and Refinement

The goal of the Image Quality Assessment and Refinement process was to obtain high-quality images from each MRI scanner that was used in the research network. Sequence refinement aimed to: (1) Maximize image resolution with minimal signal-to-noise and artifact, (2) Visualize enhancement of plaque with the shortest post-contrast wait time and a minimal amount of contrast required, and (3) Visualize structural details of lipid core and fibrous cap on T2- and T1-weighted images (pre- and post-contrast).

The Image Quality Assessment and Refinement process began at PUMC. The PUMC 3T Siemens Skyra (same model as used at MUSC) initially scanned the phantom and the images were assessed by the ChAMPION Executive Committee. The ChAMPION Neuroradiologists and Physicist made adjustments in the sequences, if required. Once sequence image quality was assured using the phantom, the sequences were tested on

approximately two normal subjects. Testing of the MRI sequences with normal human subjects was required to ensure that image degradation due to common artifacts, such as "ghosting" and blood or CSF flow artifacts, could be minimized prior to scanning ICAS patients, who may be unable to tolerate longer scan times. Images from the normal subjects were reviewed by the ChAMPION Executive Committee and additional adjustments in the sequences were made as required to obtain the necessary image quality. Once the sequences tested on normal subjects were finalized by the committee, sequence tests on ICAS patients (ideally two middle cerebral artery stenosis, two internal carotid artery stenosis, and two basilar artery stenosis) were planned.

The above process was then repeated on a second MRI scanner at PUMC (the 3T GE Discovery MR750). After the sequences were finalized for the PUMC scanners, the process was repeated at each Chinese site to assess image quality and make sequence adjustments as needed on other MRI scanner models (e.g., Phillips or GE). During the Image Quality Assessment and Refinement process, a secure shared file system at MUSC was used to store and review images.

*2.5. Collaboration and Interaction*

The principal investigators in MUSC and PUMC continued to collaborate to combine the resources of their institutions and nations. Methods of communication included Skype and WeChat video communication. Scheduled meetings, as well as ad-hoc sequence trouble-shooting discussions, were held. Topics of discussion included a selection of Chinese sites, assessment of image quality and refinement of MRI sequences, regulatory approval at sites, subject recruitment, and design of the next study. Throughout the study period, the ChAMPION study team also remained in close communication via email. The ChAMPION investigators met in person at least once per year.

## 3. Results

PUMC was successfully developed as a CCC, with a Project Manager who coordinated efforts across the sites. Case report form surveys were sent to eligible sites with multiple responses. We selected 11 sites within China that had experience utilizing VW-MR sequences and had the capacity to implement a research protocol. Those final sites are shown in Table 1. Sites were selected based on several factors including geographic location and MRI equipment capabilities.

A ChAMPION investigators' meeting was held in association with the 16th Congress of Chinese Cerebrovascular Diseases in Hangzhou, China in April 2016 after site selection. The principal investigators and Project Managers from both PUMC and MUSC were in attendance.

The ChAMPION researchers successfully developed the VW-MR research infrastructure and completed VW-MR sequence optimization, implementation, and testing at the Chinese sites. Due to differences in MRI manufacturers and software at the sites, the original VW-MR sequences were modified from the original MRI platform (Siemens) to run on other MRI scanner types (e.g., GE and Phillips, Table 2). T2 sequences were optimized for the GE and Phillips scanners and T1 sequence modification was successful. The ICAS plaque phantom had been successfully scanned at six sites in the US [14]. Of the original 11 sites in China, six sites were able to successfully load the sequences and scan the phantom (Table 3, Figure 1). Visual differences between GE and Siemens scanners were noted in the phantom images. Five Chinese sites were unable to scan the phantom due to two primary challenges: unsuccessful cooperation between the neurology and radiology departments and competing conflict with existing VW-MR research projects.

**Table 1.** Selected VW-MR sites in China.

| Site Name | How Many Patients Who Have 50–99% Stenosis | How Was the Answer Determined? | MRI Scanner Magnet | Make and Model | Software | Head Coil | Available for Research |
|---|---|---|---|---|---|---|---|
| Fujian Medical University Union Hospital | 160–180 | Registry and estimation | 3 | GE Discovery MR750 3.0T | 2.6.27 | 8 | Yes, patients |
| Peking Union Medical College Hospital | 100 | registry | 3 | Siemens Skyra, GE750 | VA20, DV24 | 32 | Yes, patients |
| Second Affiliated Hospital of Zhejiang University | 100 | estimation | 1.5 and 3 | GE 750W | DV 23.1_1317c | 8 | Yes, do not use it |
| Northern Jiangsu People's Hospital | 275 | registry | 1.5 and 3 | GE MR750; GE MR 750W: GE MR360 | GE DV 24 | 8, 16, 32 | Yes, patients |
| Daping Hospital, Third Military Medical University | 2000+ | estimation | 1.5 and 3 | Siemens Verio 3T | VB17 | 8 | Yes, Research Only |
| Chinese People's Armed Police General Hospital | 66 | estimation | 1.5 and 3 | Siemens Trio Tim | Syngo MR B17 | 8 | Yes, patients |
| The First Hospital of Jilin University | 100's | estimation | 1.5 and 3 | Siemens Trio | Siemens VB15 | 12 | Yes, patients |
| Tangshan Gongren Hospital | 40 | estimation | 1.5 and 3 | GE Signa HDst: Philips Achieva | GE Release HD16.0_1131a; Philips Release 3.2.3.2 | 8, 16 | Yes, do not use it |
| The First Affiliated Hospital of University of South China | 800 | Registry and estimation | 1.5 and 3 | Philips Achieva TX 3.0T | Achieva Release 2.6.3.6 | 8, 16 | Yes, patients |
| Peking University First Hospital | 500 | estimation | 3 | GE MR750 | DV22 | 32 | Yes, patients |
| General Hospital of Shenyang Military Region | 800–1000 | estimation | 1.5 and 3 | GE Discovery MR750 | V03.20 | 8 | Yes, patients |

**Table 2.** List of VW-MR Scans at Chinese sites.

| Site Name | MRI Scanner Type | Number of Patient Scans | Number of Phantom Scans | Number of Incomplete Patient Scans | Total Scans |
|---|---|---|---|---|---|
| Fujian Medical University Union Hospital | GE | 11 | 1 | 0 | 12 |
| Peking Union Medical College Hospital | GE, Siemens | 13 | 2 | 1 | 16 |
| Second Affiliated Hospital of Zhejiang University | GE | 2 | 1 | 1 | 4 |
| Northern Jiangsu People's Hospital | GE | 1 | 1 | 1 | 3 |

**Table 2.** *Cont*.

| Site Name | MRI Scanner Type | Number of Patient Scans | Number of Phantom Scans | Number of Incomplete Patient Scans | Total Scans |
|---|---|---|---|---|---|
| Daping Hospital of Third Military Medical University | Siemens | 3 | 1 | 0 | 4 |
| Chinese People's Armed Police General Hospital | Siemens | 6 | 1 | 1 | 8 |
| First Hospital Jilin University | Siemens | unsuccessful | | | 0 |
| Tangshan Gongren Hospital | GE, Philips | unsuccessful | | | 0 |
| The First Affiliated Hospital of University of South China | Philips | unsuccessful | | | 0 |
| Peking University First Hospital | GE | unsuccessful | | | 0 |
| General Hospital of Shenyang Military Region | GE | unsuccessful | | | 0 |
| Total | | 36 | 7 | 4 | 47 |

**Table 3.** MRI models and parameters for phantom T1-weighted scan.

| Site Name | MRI Model | Slice Thickness, mm | Resolution | Field of View | T1-Weighted | | T2-Weighted | |
|---|---|---|---|---|---|---|---|---|
| | | | | | TR, ms | TE, ms | TR, ms | TE, ms |
| Fujian Medical University Union Hospital | GE Discovery MR750 | 1.2 | 0.468 × 0.468 | 256 × 256 | 440 | 14 | 1500 | 114 |
| Peking Union Medical College Hospital | GE Discovery MR750 | 1.2 | 0.468 × 0.468 | 256 × 256 | 440 | 14 | 1500 | 109 |
| Second Affiliated Hospital of Zhejiang University | GE Discovery MR750 | 1.2 | 0.234 × 0.234 | 512 × 512 | 440 | 14 | 1500 | 114 |
| Northern Jiangsu People's Hospital | GE Discovery MR750 | 1.2 | 0.468 × 0.468 | 256 × 256 | 440 | 15 | 1500 | 107 |
| Daping Hospital of Third Military Medical University | Siemens Verio | 1.2 | 0.4 × 0.4 | 320 × 320 | 458 | 16 | 1500 | 64 |
| Chinese People's Armed Police General Hospital | Siemens Trio Tim | 1.2 | 0.4 × 0.4 | 320 × 320 | 458 | 16 | 1500 | 68 |

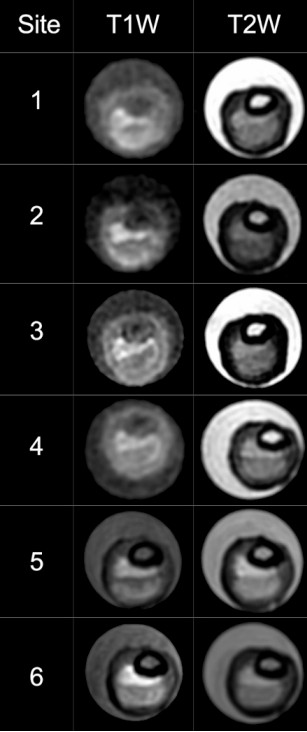

**Figure 1.** Cross-sectional T1 and T2 high-resolution MRI images of a phantom model acquired on six Chinese sites. The scans are in the same sequence of sites as in Table 3. Phantom images scanned from GE (No. 1–4) were noted to be visually different from Siemens (No. 5–6). This suggested that T1W images on GE scanners likely need further refinement. T1W: T1-weighted, T2W: T2-weighted.

After each site completed acceptable scanning of the ICAS phantom and image quality was verified by the study leadership, the site was given the approval to begin scanning normal subjects and collecting preliminary feasibility data from patients with ICAS. In total, 36 human subjects' VW-MR data were completed from the Chinese sites (Figure 2).

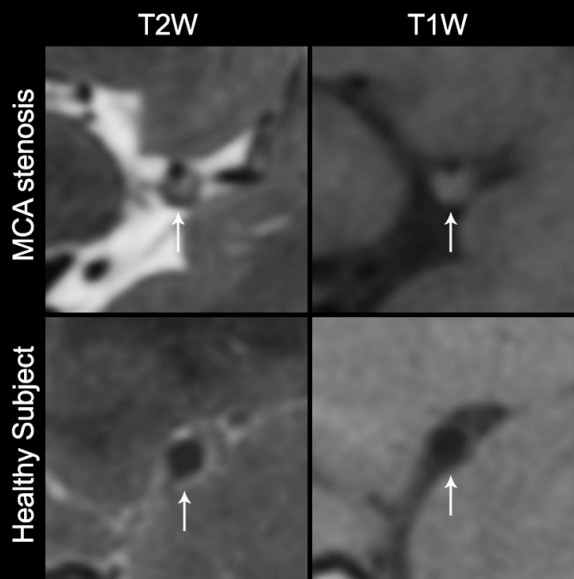

**Figure 2.** Cross-sectional non-contrast T1 and T2 high-resolution MRI of a subject with middle cerebral artery (MCA) stenosis and a healthy subject. The white arrows indicate the stenotic MCA and a thick plaque of the subject with MCA stenosis (top row) and the thin and smooth vessel wall of the healthy subject (bottom row). T1W: T1-weighted without contrast, T2W: T2-weighted.

## 4. Discussion

The ChAMPION study shows that the development of a collaborative international VW-MR research network using a disease-specific phantom is feasible. In fact, the development of the VW-MR research network in China has already helped facilitate PUMC initiating a multicenter clinical study evaluating the use of VW-MR in acute ischemic stroke [21]. The phantom was a critical component in the standardization of sequences across multiple scanner types. Using the phantom, sequences were efficiently calibrated avoiding the inherent variability present in traditional preliminary normal human subject scanning. Scanning human subjects for only the final stages of quality control made more efficient use of study resources.

We encountered several challenges while establishing this collaborative research network. One unique challenge when transitioning to new scanner models was the need to perform real-time MRI "trouble-shooting" sequence modification, which required real-time image evaluation and communication between MRI technologists and ChAMPION radiology experts. A traditional video conference was initially used but did not always have enough resolution to allow for adequate image evaluation by remote experts. The quality of the video conferencing challenge was overcome by having multiple redundant platforms available for use during scheduled meetings. The challenge in communication between MRI technologists and ChAMPION experts was overcome when the multilingual project manager was trained to have adequate knowledge of MRI and was able to travel to the test site. She was able to facilitate technical troubleshooting discussions between the site MRI technologists and the remote experts. Another challenge was the lack of scanner availability during working hours at high-volume clinical sites lacking dedicated research scanners. Radiologists and technicians working outside daytime hours reported a lack of motivation, which is a common issue encountered in clinical research. MR technologist compensation may be considered when designing similar research projects in the future. Technical difficulties with sequence installation were overcome by having experienced technologists, site radiologists, and engineers from the MR manufacturer at the scanner during the initial installation. Additional technical challenges included inconsistent scanner software versions and MR technologists modifying scanning sequence parameters manually. The creation of a scanning guide document with an abundance of figures helped facilitate MR technologist education for more consistent adherence to the required technical parameters. The document was continuously modified as additional technical challenges were discovered. Additional challenges included a 12 h time zone difference, limited access to international cloud-based data-sharing platforms, as well as regulatory and financial issues that have been reported in previous literature [22–24]. Despite these challenges, most of the Chinese sites were able to successfully scan the phantom and subjects using the standardized sequences.

In the future, such MRI networks can provide researchers with access to larger and more diverse populations to study intracranial atherosclerotic plaque, as well as other cerebrovascular pathologies. The use of disease-specific phantoms facilitates more efficient calibration of sequences between sites and scanners. Given that imaging texture differs between MRI manufacturers, the disease-specific phantoms could also lead to future research in the development of image processing algorithms for automated calibration of images acquired on different scanners, possibly utilizing machine learning techniques. The ChAMPION network could also be expanded to other countries that have a high prevalence of stroke due to ICAS, such as Southeast Asian countries, India, Pakistan, Middle Eastern countries, and European countries. This expansion would also provide the opportunity to study ethnic differences in the pathophysiology of ICAS and its response to therapies in order to further maximize stroke prevention in all parts of the world.

## 5. Conclusions

This study demonstrated the feasibility of establishing a collaborative international VW-MR research network. This is a prerequisite for achieving the scale necessary for

effective clinical trials studying potential biomarkers and treatments in many diseases including ICAS.

**Supplementary Materials:** The following supporting information can be downloaded at: https: //www.mdpi.com/article/10.3390/ctn6030016/s1, Table S1: ChAMPION Site Qualification Case Report Form.

**Author Contributions:** T.N.T. and W.-H.X. conceptualized and designed the study, obtained study funding, drafted and provided critical revision of the manuscript. Y.Y. coordinated participating Chinese centers, collected data, created figures, and drafted the manuscript. W.-H.X., M.Y. and M.-L.L. organized and coordinated participating Chinese centers, and collected data. A.R.C., T.B., M.V.S., R.M., C.D. and M.I.C. assisted with the study design and provided critical revision of manuscripts. T.L. coordinated the US and Chinese centers, created tables, and collected data. ChAMPION investigators collected data. All authors have read and agreed to the published version of the manuscript.

**Funding:** This research was funded by the National Institute of Health, Fogarty International Center, grant number 5R21TW010356.

**Institutional Review Board Statement:** Human subjects research at Chinese sites was overseen by Dr. Weihai Xu, who ensured the protection of human subjects from research risks by monitoring the regulatory process at each site according to international Human Subjects regulations.

**Informed Consent Statement:** Informed consent was obtained from all subjects involved in the study.

**Data Availability Statement:** Not applicable.

**Acknowledgments:** The authors wish to thank the following ChAMPION site leaders and contributors: Lawrence Wong, Edward Feldmann, Rick Swartz, Sameer Ansari, Timothy J Carroll, Matthew Gounis, Ju-Yu Chueh, Kajo Van Der Marel, Zoran Rumboldt, Amanda K Buck, Xiaohong Joe Zhou, Robert M King, Hui Mao, Shaokuan Zheng, Olivia W Brooks, Jeff W Pappleye, Zhangyu Zou, Min Lou, Jun Xu, Meng Zhang, and Shiwen Wu.

**Conflicts of Interest:** The following authors received salary support from NIH Fogarty International Center (R21TW010356) for their work on this project: WX, ARC, TL, TB, TNT.

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
