# Peer review of "Implementation of an International Vessel Wall MR Plaque Imaging Research Network: Experience with the ChAMPION Study"

_ctn, doi:10.3390/ctn6030016_

Round 1

Reviewer 1 Report

The study looks into a network using vessel wall imaging and found that 6 out of 11 were able to perform it correctly

Author Response

We appreciate the review and comment.

Reviewer 2 Report

In this article, Yu et al describe their experience establishing the research apparatus required for beginning larger scale investigations in China specific to vessel wall imaging in ICAS. There is no doubt that a framework for conducting further research will be useful, but it is unclear to me if the mere establishing of a research network with a methodology previously utilized at MUSC in a different locality merits publication. 

Specific comments 

  • I'm not sure it is very helpful to include selected sites that were not able to participate in the scanning of patients, and therefore unable to contribute to the established research network, in table 1. Simply stating that they were not able to do so would be easier.
  • In line 238 you discuss lack of motivation of the technologists and ancillary staff, finding individuals to conduct the research you wish to do is a problem shared across just about any any clinical study and is hardly unique
  • What happened to the one site that was attempted to be placed into the study? (line 195) If it is unsuccessful I would remove this line
  • There is no discussion on how the team plans to deal with the obvious differences in scan quality in figure 1, in fact the image quality seems radically different

Author Response

1. I'm not sure it is very helpful to include selected sites that were not able to participate in the scanning of patients, and therefore unable to contribute to the established research network, in table 1. Simply stating that they were not able to do so would be easier.

Reply:

Thank you for your comment. We have adjusted table 1 so that it has the same order as Table 2 for easier comparison. We kept the information in table 1 to show that factors such as fewer case numbers or MRI availability were not related to unsuccessful scanning.

2. In line 238 you discuss lack of motivation of the technologists and ancillary staff, finding individuals to conduct the research you wish to do is a problem shared across just about any any clinical study and is hardly unique. 

Reply: 

Thank you. We added in line 242 that this is a common issue in clinical research.

3. What happened to the one site that was attempted to be placed into the study? (line 195) If it is unsuccessful I would remove this line.

Reply:

Thank you. We have removed this sentence.

4. There is no discussion on how the team plans to deal with the obvious differences in scan quality in figure 1, in fact the image quality seems radically different

Reply:

Thank you. this is a good point,  it is a known phenomenon that different MRI manufacturers tend to create quite a different image texture. it may be helpful to utilize machine learning algorithms to adjust the image texture, making those images more comparable between different scanners.

We have added in line 259: 

It is a known phenomenon that the texture difference between MRI manufacturers, the disease-specific phantoms could also lead to future research in the development of image processing algorithms for automated calibration of images acquired on different scanners, possibly utilizing machine learning techniques. 

Reviewer 3 Report

Excellent study that shows that development of a collaborative international VW-216 MR research network using a disease-specific phantom which could be feasible. excellent described the methods and I don't see any issues.

Author Response

We appreciate your review and comments.

Round 2

Reviewer 2 Report

The authors have addressed some of my comments sufficiently